# A Boosting-Type Convergence Result for ADABOOST.MH with Factorized Multi-Class Classifiers

Xin Zou[*]  Zhengyu Zhou[*]  Jingyuan Xu  Weiwei Liu[†]

School of Computer Science, Wuhan University
National Engineering Research Center for Multimedia Software, Wuhan University
Institute of Artificial Intelligence, Wuhan University
Hubei Key Laboratory of Multimedia and Network Communication Engineering, Wuhan University
{zouxin2021, zzysince1999, jingyuanxu777, liuweiwei863}@gmail.com

## Abstract

ADABOOST is a well-known algorithm in boosting. Schapire and Singer propose, an extension of ADABOOST, named ADABOOST.MH, for multi-class classification problems. Kégl shows empirically that ADABOOST.MH works better when the classical one-against-all base classifiers are replaced by factorized base classifiers containing a binary classifier and a vote (or code) vector. However, the factorization makes it much more difficult to provide a convergence result for the factorized version of ADABOOST.MH. Then, Kégl raises an open problem in COLT 2014 to look for a convergence result for the factorized ADABOOST.MH. In this work, we resolve this open problem by presenting a convergence result for ADABOOST.MH with factorized multi-class classifiers.

## 1 Introduction

Boosting is an approach to machine learning based on the idea of creating a highly accurate prediction rule by combining many relatively weak and inaccurate rules [19] and has inspired a lot on theoretical analysis and algorithm design in supervised learning [11, 17]. The seminal algorithm in boosting, ADABOOST [9], requires no knowledge of the upper bound of the edge, which makes it convenient in practice.

In addition to to binary ADABOOST, [9] also proposes two multi-class extensions, named AD-ABOOST.M1 and ADABOOST.M2. Then, Schapire and Singer's seminal paper [20] proposes another extension named ADABOOST.MH. The main idea of ADABOOST.MH is to use vector-valued base classifiers to build a multi-class discriminant function of $K$ outputs when there are $K$ classes, and then replace the weight vector in ADABOOST with a weight matrix over instances and labels.

The simplest implementation of the concept in ADABOOST.MH is to use $K$ independent one-against-all classifiers in which base classifiers are only loosely connected through the common normalization of the weight matrix. However, [15] points out that such an implement is suboptimal in most of the practical problems since it is limited to only decision stumps weak learners. To solve this problem, [15] proposes another base learner named multi-class Hamming trees, which optimizes the multi-class edge without reducing the problem to $K$ binary classifications. The key idea in [15] is to factorize general vector-valued classifiers $\mathbf{h}$ into an input-independent code vector of length $K$, i.e., $\mathbf{v} \in \{-1, +1\}^K$, and label-independent scalar classifier $\varphi$. However, [15] gets in trouble when proving the convergence rate of the proposed implement of ADABOOST.MH due to the factorization

---

[*]equal contribution

[†]Corresponding author: Weiwei Liu (liuweiwei863@gmail.com).

38th Conference on Neural Information Processing Systems (NeurIPS 2024).

step. So [14] raises an open problem in COLT 2014, looking for a convergence rate of the factorized ADABOOST.MH in [15], with limited dependence on the sample size $n$.

Our **contributions** can be concluded as follows:

1. We provide a convergence result (Theorem 3.3) of factorized ADABOOST.MH, where the step $T^*$ which guarantees the training error to be 0 is of order $O\left(n^2 \ln n\right)$.

2. According to the requirement of [14], we improve the dependence on $n$ and resolve the open problem by providing a convergence result (Theorem 3.4) where $T^*$ is of order $O\left(K \ln(nK)\right)$. This result greatly improves when $n$ is much larger than $K$.

More related works are deferred to Appendix B.

## 2 Preliminaries

We consider a multi-class classification problem where the input space is $\mathcal{X} = \mathbb{R}^d$ and $\mathcal{L} = [K]$ is the label space, where $K$ is the number of classes and $[K] := \{1, \ldots, K\}$. Assume we attain the training data $\mathcal{D}_L = \{(\mathbf{x}_1, \ell(\mathbf{x}_1)), \ldots, (\mathbf{x_n}, \ell(\mathbf{x}_n))\}$, where $\ell(\mathbf{x}_i) \in \mathcal{L}$ is the label of $\mathbf{x}_i$. Since we want to use vector-valued classifiers, it is convenient to use the one-hot labels $\mathbf{y}_i \in \{-1, +1\}^K$ for $\mathbf{x}_i$, where $\mathbf{y}_i(\ell(\mathbf{x}_i)) = 1$ and all the other elements are $-1$. We use the new dataset $\mathcal{D} = \{(\mathbf{x}_1, \mathbf{y}_1), \ldots, (\mathbf{x}_n, \mathbf{y}_n)\}$ as the input data of ADABOOST.MH and define an observation matrix $\mathbf{X} := (\mathbf{x}_1, \ldots, \mathbf{x}_n)^T \in \mathbb{R}^{n \times d}$, a label matrix $\mathbf{Y} := (\mathbf{y}_1, \ldots, \mathbf{y}_n)^T \in \{-1, +1\}^{n \times K}$. We call $\mathbf{y}$ and $\ell$ the label and the index of $\mathbf{x}$ respectively, as in [14].

[14] considers a special case of ADABOOST.MH, where each weak classifier has a specialized structure. ADABOOST.MH returns a vector-valued discriminant function $\mathbf{f} : \mathcal{X} \to \mathbb{R}^K$ with a combined predictor $\mathbf{F_f} : \mathcal{X} \to \{-1, +1\}^K$ where $\mathbf{F_f}(\mathbf{x})_l = \text{sign}(\mathbf{f}(\mathbf{x})_l)$ for $l = 1, \ldots, K$. Here and in this paper, we define

$$\text{sign}(x) = \begin{cases} +1 & x \geq 0 \\ -1 & x < 0 \end{cases}.$$

The goal of the ADABOOST.MH algorithm [20] is to return $\mathbf{f}$ such that the Hamming loss of $\mathbf{F_f}$,

$$\widehat{R}_{\text{H}}(\mathbf{F_f}, \mathbf{W}) := \sum_{i=1}^{n} \sum_{l=1}^{K} w_{i,l} \mathbb{I}\{\mathbf{F_f}(\mathbf{x}_i)_l \neq y_{i,l}\}, \tag{1}$$

is as small as possible, where $\mathbb{I}(\cdot)$ is the indicator function and $\mathbf{W} = [w_{i,l}] \in [0, 1]^{n \times K}$ is a distribution over the data points and the labels. $\mathbf{W}$ can be chosen as any distribution over $[n] \times [K]$ and is different in different papers. In [20], the authors set $w_{i,l} = \frac{1}{nK}$ for any $i \in [n], l \in [K]$. Here, we follow [14] and set

$$w_{i,l} = \begin{cases} \frac{1}{2n} & \text{if } y_{i,l} = +1 \\ \frac{1}{2n(K-1)} & \text{if } y_{i,l} = -1 \end{cases}. \tag{2}$$

We define the weighted multi-class exponential margin-based error

$$\widehat{R}_{\text{EXP}}(\mathbf{f}, \mathbf{W}) := \sum_{i=1}^{n} \sum_{l=1}^{K} w_{i,l} \exp\left(-\mathbf{f}(\mathbf{x}_i)_l \cdot y_{i,l}\right) \tag{3}$$

as a surrogate for $\widehat{R}_{\text{H}}(\mathbf{F_f}, \mathbf{W})$. Since $\mathbb{I}\{\mathbf{F_f}(\mathbf{x}_i)_l \neq y_{i,l}\} = \mathbb{I}\{\mathbf{f}(\mathbf{x}_i)_l \cdot y_{i,l} \leq 0\} \leq \exp\left(-\mathbf{f}(\mathbf{x}_i)_l \cdot y_{i,l}\right)$, we can get that $\widehat{R}_{\text{H}}(\mathbf{F_f}, \mathbf{W}) \leq \widehat{R}_{\text{EXP}}(\mathbf{f}, \mathbf{W})$.

It's well-known that ADABOOST directly minimizes the exponential loss [19, Chapter 7], then, we can apply the ADABOOST algorithm to the extended binary training set $\cup_{i=1}^{n} \{(\mathbf{x}_i, y_{i,l})\}_{l=1}^{K}$, yielding the ADABOOST.MH algorithm, which directly minimizes $\widehat{R}_{\text{EXP}}(\mathbf{f}, \mathbf{W})$ and output the final discriminant function $\mathbf{f}^{(T)}(\cdot)$, where $\mathbf{f}^{(T)}(\mathbf{x}) = \sum_{t=1}^{T} \mathbf{h}^{(T)}(\mathbf{x})$ is a sum of $T$ base classifiers $\mathbf{h}^{(t)} : \mathbb{R}^d \to \mathbb{R}^K$ returned by a base learner algorithm BASE$(\mathbf{X}, \mathbf{Y}, \mathbf{W}^{(t)})$ in each iteration $t$.

Define

$$Z(\mathbf{h}, \mathbf{W}) = \sum_{i=1}^{n} \sum_{l=1}^{K} w_{i,l} \exp\left(-\mathbf{h}(\mathbf{x}_i)_l \cdot y_{i,l}\right), \tag{4}$$

by a similar calculation in [19, Proof of Theorem 3.1], we can obtain that:

$$\widehat{R}_{\text{EXP}}(\mathbf{f}^{(T)}, \mathbf{W}) = \prod_{t=1}^{T} Z(\mathbf{h}^{(t)}, \mathbf{W}^{(t)}).$$

According to the above discussion, we know that to minimize $\widehat{R}_{\text{EXP}}(\mathbf{f}^{(T)}, \mathbf{W})$, the base learner needs to find a $\mathbf{h}^{(t)}$ that minimizes $Z(\mathbf{h}^{(t)}, \mathbf{W}^{(t)})$ at the $t$-th iteration. In the following, we introduce two choices of $\mathbf{h}$ in [20] and [15], the corresponding convergence rate of $\widehat{R}_{\text{EXP}}(\mathbf{f}^{(T)}, \mathbf{W})$, and problems when trying to get a convergence rate of $\widehat{R}_{\text{EXP}}(\mathbf{f}^{(T)}, \mathbf{W})$ for factorized ADABOOST.MH.

## 2.1  Unfactorized Choice

[20] considers using $\mathbf{h}$ with the form $\mathbf{h}(\mathbf{x}) = \alpha\boldsymbol{\varphi}(\mathbf{x})$, where $\alpha \in \mathbb{R}$ and $\boldsymbol{\varphi} : \mathbb{R}^d \to \{-1, +1\}^K$ can be seen as the vector consists of $K$ binary classifiers $\varphi_1, \ldots, \varphi_K$.

We consider the $t$-th iteration, and to simplify the notations, we omit the superscript $t$ and use $\mathbf{W}, \mathbf{h}, \boldsymbol{\varphi}, \alpha$ to represent $\mathbf{W}^{(t)}, \mathbf{h}^{(t)}, \boldsymbol{\varphi}^{(t)}, \alpha^{(t)}$ respectively. According to [20], if we define

$$r = \sum_{i=1}^{n} \sum_{l=1}^{K} w_{i,l} \cdot y_{i,l} \cdot \boldsymbol{\varphi}(\mathbf{x}_i)_l \tag{5}$$

as the edge, then we have

$$Z(\mathbf{h}, \mathbf{W}) = \sum_{i=1}^{n} \sum_{l=1}^{K} w_{i,l} \exp\left(-\mathbf{h}(\mathbf{x}_i)_l \cdot y_{i,l}\right) = \sum_{i=1}^{n} \sum_{l=1}^{K} w_{i,l} \exp\left(-\alpha\boldsymbol{\varphi}(\mathbf{x}_i)_l \cdot y_{i,l}\right)$$

$$= \sum_{i,l:\boldsymbol{\varphi}(\mathbf{x}_i)_l \cdot y_{i,l}=1} w_{i,l} \cdot e^{-\alpha} + \sum_{i,l:\boldsymbol{\varphi}(\mathbf{x}_i)_l \cdot y_{i,l}=-1} w_{i,l} \cdot e^{\alpha}.$$

Since $\sum_{i,l:\boldsymbol{\varphi}(\mathbf{x}_i)_l \cdot y_{i,l}=1} w_{i,l} + \sum_{i,l:\boldsymbol{\varphi}(\mathbf{x}_i)_l \cdot y_{i,l}=-1} w_{i,l} = 1$ and $\sum_{i,l:\boldsymbol{\varphi}(\mathbf{x}_i)_l \cdot y_{i,l}=1} w_{i,l} - \sum_{i,l:\boldsymbol{\varphi}(\mathbf{x}_i)_l \cdot y_{i,l}=-1} w_{i,l} = r$, we can get that

$$\sum_{i,l:\boldsymbol{\varphi}(\mathbf{x}_i)_l \cdot y_{i,l}=1} w_{i,l} = \frac{1+r}{2}, \qquad \sum_{i,l:\boldsymbol{\varphi}(\mathbf{x}_i)_l \cdot y_{i,l}=-1} w_{i,l} = \frac{1-r}{2}.$$

So we have:

$$Z(\mathbf{h}, \mathbf{W}) = \frac{1+r}{2} \cdot e^{-\alpha} + \frac{1-r}{2} \cdot e^{\alpha}.$$

Fix $\boldsymbol{\varphi}$ first, minimizing $Z(\mathbf{h}, \mathbf{W})$ over $\alpha$ yields that:

$$\alpha = \frac{1}{2} \ln\left(\frac{1+r}{1-r}\right).$$

This gives

$$Z(\mathbf{h}, \mathbf{W}) = \sqrt{1 - r^2}.$$

Then choose $\boldsymbol{\varphi}$ to minimize $\sqrt{1 - r^2}$, i.e., maximize $|r|$. If we have $r^{(t)} \geq \delta > 0$ for all $t$, then we can get:

$$\widehat{R}_{\text{EXP}}(\mathbf{f}^{(T)}, \mathbf{W}) = \prod_{t=1}^{T} \sqrt{1 - \left(r^{(t)}\right)^2} \leq \left(\sqrt{1 - \delta^2}\right)^T \leq \exp\left(-\frac{\delta^2}{2}T\right),$$

which means that the weighted exponential error goes to error exponentially fast. Let $\exp\left(-\frac{\delta^2}{2}T\right) < \frac{1}{nK}$, we know that the weighted Hamming error becomes zero after

$$T^* = \left\lceil \frac{2\ln(nK)}{\delta^2} \right\rceil + 1$$

iterations. The condition $r^{(t)} \geq \delta > 0$ for all $t$ is satisfied when the empirically weak learning condition on the classifier $\boldsymbol{\varphi}$ holds for the extended binary training set $\cup_{i=1}^{n}\{(\mathbf{x}_i, y_{i,l})\}_{l=1}^{K}$.

**Definition 2.1** (empirically $\delta$-weak learning condition). For a given binary dataset $\{(\mathbf{x}_1, y_1), \ldots, (\mathbf{x}_m, y_m)\}$ where $y_i \in \{-1, +1\}$, we say that the empirically $\delta$-weak learning condition holds for some $\delta > 0$ if for any distribution $\mathbf{w} \in \Delta^{m-1}$ over $[m]$, we can always find a binary classifier $\varphi : \mathcal{X} \to \{-1, +1\}$ such that:

$$\gamma = \sum_{i=1}^{m} \mathbf{w}_i \cdot y_i \cdot \varphi(\mathbf{x}_i) \geq \delta,$$

where

$$\Delta^{m-1} = \left\{ \boldsymbol{\lambda} \in \mathbb{R}^m \middle| \boldsymbol{\lambda}_i \geq 0 \; \forall i \in [m], \sum_{i=1}^{m} \boldsymbol{\lambda}_i = 1 \right\}$$

is the $(m-1)$-dimensional probability simplex.

## 2.2 Factorized Choice

The original ADABOOST.MH [20] reduces the multi-class problem into $K$ binary one-against-all classifications. [15] avoids such a reduction by factorizing the vector-valued classifier $\mathbf{h}$ into an input-independent vector of length $K$ and a label-independent scalar classifier. Formally, [15] sets

$$\mathbf{h}(\mathbf{x}) = \alpha \mathbf{v} \varphi(\mathbf{x}),$$

where $\alpha \in \mathbb{R}^+$ is a positive real-valued base coefficient, $\mathbf{v} \in \{-1, +1\}^K$ is an input-independent vote (or code) vector of length $K$, and $\varphi : \mathbb{R}^d \to \{-1, +1\}$ is a label-independent binary classifier. For more details about the factorized ADABOOST.MH, please refer to Algorithm 1 in Appendix A.

We consider the $t$-th iteration, and to simplify the notations, we omit the superscript $t$ and use $\mathbf{W}, \mathbf{h}, \varphi, \alpha, \mathbf{v}$ to represent $\mathbf{W}^{(t)}, \mathbf{h}^{(t)}, \varphi^{(t)}, \alpha^{(t)}, \mathbf{v}^{(t)}$ respectively. [15] shows that

$$Z(\mathbf{h}, \mathbf{W}) = \frac{e^\alpha + e^{-\alpha}}{2} - \frac{e^\alpha - e^{-\alpha}}{2} \cdot \sum_{l=1}^{K} v_l \left( \mu_{l+} - \mu_{l-} \right),$$

where

$$\mu_{l-} = \sum_{i=1}^{n} w_{i,l} \mathbb{I}\{\varphi(\mathbf{x}_i) \neq y_{i,l}\}$$

is the weighted per-class error rate and

$$\mu_{l+} = \sum_{i=1}^{n} w_{i,l} \mathbb{I}\{\varphi(\mathbf{x}_i) = y_{i,l}\}$$

is the weighted per-class correct classification rate for each class $l = 1, \ldots, K$. Similar to Equation (5), we define the multi-class edge of the classifier $\mathbf{h}$ as

$$\gamma = \gamma(\mathbf{v}, \varphi, \mathbf{W}) = \sum_{l=1}^{K} \gamma_l = \sum_{l=1}^{K} v_l \left( \mu_{l+} - \mu_{l-} \right) = \sum_{i=1}^{n} \varphi(\mathbf{x}_i) \sum_{l=1}^{K} w_{i,l} \cdot v_l \cdot y_{i,l}, \tag{6}$$

where

$$\gamma_l = v_l \left( \mu_{l+} - \mu_{l-} \right) = \sum_{i=1}^{n} w_{i,l} \cdot v_l \cdot \varphi(\mathbf{x}_i) \cdot y_{i,l}$$

is the classwise edge of $\mathbf{h}$. By a similar calculation as in Section 2.1, we know that $Z(\mathbf{h}, \mathbf{W})$ is minimized when we set

$$\alpha = \frac{1}{2} \ln \left( \frac{1 + \gamma}{1 - \gamma} \right),$$

which gives

$$Z(\mathbf{h}, \mathbf{W}) = \sqrt{1 - \gamma^2}.$$

So in order to minimize $Z(\mathbf{h}, \mathbf{W})$, we need to choose $\mathbf{v}$ and $\varphi$ to maximize $|\gamma|$. From the equation $\gamma(\mathbf{v}, \varphi, \mathbf{W}) = \sum_{l=1}^{K} v_l \left( \mu_{l+} - \mu_{l-} \right)$, we know that if $\gamma(\mathbf{v}, \varphi, \mathbf{W}) \leq 0$, then $\gamma(-\mathbf{v}, \varphi, \mathbf{W}) =$

$-\gamma(\mathbf{v}, \varphi, \mathbf{W}) \geq 0$. So the problem reduces to finding $\mathbf{v}, \varphi$ that maximize $\gamma$. From Equation (6) we know that for fixed $\varphi$, $\gamma$ is maximized when we choose $\mathbf{v}$ as

$$v_l = \begin{cases} +1 & \mu_{l+} \geq \mu_{l-} \\ -1 & \mu_{l+} < \mu_{l-} \end{cases} \tag{7}$$

for all classes $l = 1, \ldots, K$.

Similar to Section 2.1, if there exists a number $\delta > 0$ such that $\gamma\left(\mathbf{v}^{(t)}, \varphi^{(t)}, \mathbf{W}^{(t)}\right) \geq \delta$ for all $t = 1, \ldots, T$, then we can get an upper bound for $\widehat{R}_{\mathrm{EXP}}(\mathbf{f}^{(T)}, \mathbf{W})$:

$$\widehat{R}_{\mathrm{EXP}}(\mathbf{f}^{(T)}, \mathbf{W}) = \prod_{t=1}^{T} \sqrt{1 - \gamma\left(\mathbf{v}^{(t)}, \varphi^{(t)}, \mathbf{W}^{(t)}\right)^2} \leq \left(\sqrt{1 - \delta^2}\right)^T \leq \exp\left(-\frac{\delta^2}{2}T\right),$$

which means that the weighted exponential error goes to error exponentially fast. Let $\exp\left(-\frac{\delta^2}{2}T\right) < \frac{1}{2n(K-1)}$, we know that the weighted Hamming error becomes zero after

$$T^* = \left\lceil \frac{2\ln(2n(K-1))}{\delta^2} \right\rceil + 1$$

iterations. To get the exponential convergence rate, the question now is whether there exists a number $\delta > 0$ such that $\gamma\left(\mathbf{v}^{(t)}, \varphi^{(t)}, \mathbf{W}^{(t)}\right) \geq \delta$ for all $t = 1, \ldots, T$.

## 2.3 Conditions for the Two Choices

For the condition in the unfactorized choice, if the empirically $\delta'$-weak learning condition holds, then for a fixed weight matrix $\mathbf{W}$, let $I = \{l \in [K] \big| \sum_{i=1}^{n} w_{i,l} > 0\}$, then for all $l \in I$, there exists a binary classifier $\varphi_l$ such that

$$r_l = \sum_{i=1}^{n} \frac{w_{i,l}}{\sum_{i=1}^{n} w_{i,l}} \varphi_l(\mathbf{x}_i) y_{i,l} \geq \delta',$$

then we can find a $\varphi$ such that $\varphi_l = \varphi_l$ for $l \in I$ so that

$$r = \sum_{i=1}^{n} \sum_{l=1}^{K} w_{i,l} \cdot \varphi_l(\mathbf{x}_i) \cdot y_{i,l} = \sum_{l \in I} \sum_{i=1}^{n} w_{i,l} \cdot \varphi_l(\mathbf{x}_i) \cdot y_{i,l} \geq \sum_{l \in I} \sum_{i=1}^{n} w_{i,l} \cdot \delta' = \delta'.$$

So the empirically $\delta'$-weak learning condition is sufficient for an exponential convergence rate for the ADABOOST.MH algorithm in [20].

For the factorized choice proposed in [15], we can not use the above argument since $\mathbf{h}$ is factorized and we need to find a binary classifier $\varphi$ for all $l = 1, \ldots, K$, while for the unfactorized choice, we can find $K$ binary classifiers $\varphi_1, \ldots, \varphi_K$ separately for each class. In [14], the author tries to solve this problem by constructing pseudo-weights and pseudo-labels and then applying the empirically $\delta'$-weak learning condition to the constructed dataset $\{(\mathbf{x}_1, y_1'), \ldots, (\mathbf{x}_n, y_n')\}$.

[14] rewrites $\gamma$ as

$$\gamma = \sum_{i=1}^{n} \varphi(\mathbf{x}_i) \sum_{l=1}^{K} w_{i,l} \cdot v_l \cdot y_{i,l} = \sum_{i=1}^{n} \varphi(\mathbf{x}_i) \sum_{l=1}^{K} w_{i,l} \left[\mathbb{I}\{v_l \cdot y_{i,l} = +1\} - \mathbb{I}\{v_l \cdot y_{i,l} = -1\}\right]$$

$$= \sum_{i=1}^{n} \varphi(\mathbf{x}_i)(w_i^+ - w_i^-) = \sum_{i=1}^{n} \varphi(\mathbf{x}_i)\mathrm{sign}(w_i^+ - w_i^-)|w_i^+ - w_i^-|,$$

where we define

$$w_i^+ = \sum_{l=1}^{K} w_{i,l}\mathbb{I}\{v_l \cdot y_{i,l} = +1\}, \quad w_i^- = \sum_{l=1}^{K} w_{i,l}\mathbb{I}\{v_l \cdot y_{i,l} = -1\}$$

for simplicity. Then we define $y_i' = \mathrm{sign}(w_i^+ - w_i^-)$ as the $i$-th pseudo-label and $w_i' = |w_i^+ - w_i^-|$ as the $i$-th pseudo-weight, then

$$\gamma = \sum_{i=1}^{n} w_i' \cdot y_i' \cdot \varphi(\mathbf{x}_i).$$

However, since $\sum_{i=1}^{n} w_i' = \sum_{i=1}^{n} |w_i^+ - w_i^-| \le \sum_{i=1}^{n}(w_i^+ + w_i^-) = 1$, $\mathbf{w}' = (w_1', \ldots, w_n')$ is not necessarily a distribution on $[n]$. To make use of the empirically $\delta'$-weak learning condition, we define

$$w_\Sigma' := \sum_{i=1}^{n} w_i' \le 1.$$

If we can get a lower bound $\omega > 0$ such that $w_\Sigma' \ge \omega$, then we have:

$$\gamma = \sum_{i=1}^{n} w_i' \cdot y_i' \cdot \varphi(\mathbf{x}_i) = w_\Sigma' \sum_{i=1}^{n} \frac{w_i'}{w_\Sigma'} \cdot y_i' \cdot \varphi(\mathbf{x}_i) \ge w_\Sigma' \cdot \delta' \ge \omega \cdot \delta',$$

where the first inequality is from the empirically $\delta'$-weak learning condition. Since the number of examples $n$ may be very large, we wish the lower bound $\omega$ to be independent of $n$, but it can depend on the number of classes $K$.

Then [14] raises an **open problem**:

> *Whether there exists a setup $(\mathbf{X}, \mathbf{W}, \mathbf{Y}$, and function class) in which all of the $2^K$ different vote vectors $\mathbf{v} \in \{-1, +1\}^K$ lead to arbitrarily small (or zero) $w_\Sigma'$, or we can find a constant (independent of $n$) lower bound $\omega$ such that with at least one vote vector and classifier $\varphi$, $w_\Sigma' \ge \omega$ holds?*

We resolve this open problem by showing that:

> *There exists a constant $\omega = \frac{1}{\sqrt{2K}}$ such that: for any $\mathbf{X}, \mathbf{W}, \mathbf{Y}$ and function class, there always exists a vote vector $\mathbf{v}$ s.t. $w_\Sigma' \ge \omega$ holds. With this result, if the empirically $\delta'$-weak learning condition holds, then for any $\mathbf{X}, \mathbf{W}, \mathbf{Y}$, there always exists a vote vector $\mathbf{v}$ and a binary classifier $\varphi$ such that $\gamma = \sum_{i=1}^{n} \varphi(\mathbf{x}_i) \sum_{l=1}^{K} w_{i,l} \cdot v_l \cdot y_{i,l} \ge \frac{\delta'}{\sqrt{2K}}$. So if we run the AD-ABOOST.MH algorithm with factorized $\mathbf{h}$, $\widehat{R}_{EXP}(\mathbf{f}^{(T)}, \mathbf{W})$ becomes zero after at most $T^* = \left\lceil \frac{4K \ln(2n(K-1))}{(\delta')^2} \right\rceil + 1$ iterations.*

## 3 Our Solution

In this section, we provide formal theorems for our above answer to the open problem and further discussions.

Because the training set size $n$ may be very large, [15] requires the lower bound to be independent of the training set size $n$ (but can be dependent on the number of classes $K$), which is much more difficult than finding a lower bound depends on $n$. To consider this problem more holistically, we provide two lower bounds, one depends on $n$ and another depends on $K$.

To solve this problem, we first formulate the problem of "finding a constant $\omega$ such that for any training set and weight matrix, there exists a code vector $\mathbf{v}$ such that $w_\Sigma' \ge \omega$ ($w_\Sigma'$ depends on the training set, weight matrix, and the code vector)" into "finding the lower bound of a constrained minimax problem". We then provide a $n$-dependent lower bound by the fact $\|\mathbf{x}\|_1 \ge \|\mathbf{x}\|_\infty$ and the fact that the maximum is not smaller than the average, where $\|\cdot\|_p$ is the $\ell_p$-norm of a vector. For the $n$-independent lower bound, we choose to lower bound the expected value of $w_\Sigma'$ when the code vector $\mathbf{v}$ is drawn from some distribution $\mathcal{D}$ on $\{-1, +1\}^K$. To eliminate the trouble caused by the labels, we choose $\mathbf{v}$ to be a Rademacher random vector with independent elements, i.e., $\mathbf{v} = (\varepsilon_1, \ldots, \varepsilon_K)$ where $\mathbb{P}[\varepsilon_i = 1] = \mathbb{P}[\varepsilon_i = -1] = \frac{1}{2}$ for $i = 1, \ldots, K$. We then provide the lower bound with the help of Khintchine inequality [10].

We define

$$\mathcal{W} := \left\{ \mathbf{W} \in \mathbb{R}^{n \times K} \,\middle|\, \mathbf{W}_{i,l} \geq 0 \text{ for all } i \in [n], l \in [K]; \sum_{i=1}^{n} \sum_{l=1}^{K} \mathbf{W}_{i,l} = 1 \right\}$$

as the set of all possible $\mathbf{W}$. Let $\mathbf{e}(\cdot) : [K] \to \{-1, +1\}^K$ be

$$\mathbf{e}(l)_i = \begin{cases} +1 & i = l \\ -1 & i \neq l \end{cases},$$

we the define $\mathcal{Y} := \left\{ (\mathbf{e}(l_1), \ldots, \mathbf{e}(l_n))^T \in \{-1, +1\}^{n \times K} | l_1, \ldots, l_n \in \mathcal{L} = [K] \right\}$ as the set of all possible $\mathbf{Y}$, and define $\mathcal{V} = \{-1, +1\}^K$ as the set of all possible $\mathbf{v}$. We then have:

$$\begin{aligned}
w'_\Sigma(\mathbf{W}, \mathbf{Y}, \mathbf{v}) &= \sum_{i=1}^{n} |w_i^+ - w_i^-| \\
&= \sum_{i=1}^{n} \left| \sum_{l=1}^{K} w_{i,l} \left\{ \mathbb{I}[v_l y_{i,l} = +1] - \mathbb{I}[v_l y_{i,l} = -1] \right\} \right| \\
&= \sum_{i=1}^{n} \left| \sum_{l=1}^{K} w_{i,l} \cdot v_l \cdot y_{i,l} \right| \\
&= \sum_{i=1}^{n} \left| \langle (\mathbf{W} \odot \mathbf{Y})_i^T, \mathbf{v} \rangle \right| = \| (\mathbf{W} \odot \mathbf{Y}) \cdot \mathbf{v} \|_1,
\end{aligned}$$

where $\odot$ is the Schur product and $\|\mathbf{x}\|_1 = \sum_{i=1}^{n} |x_i|$ is the $\ell_1$-norm of the vector $\mathbf{x}$.

The following two facts translate the problem that we are concerned with into a minimax problem.

**Fact 3.1.** The following two statements are equivalent:

(1) There exists a setup $(\mathbf{X}, \mathbf{W}, \mathbf{Y})$ in which all of the $2^K$ different vote vectors $\mathbf{v} \in \mathcal{V}$ lead to arbitrarily small (or zero) $w'_\Sigma$.

(2) $\min\limits_{\mathbf{W} \in \mathcal{W}, \mathbf{Y} \in \mathcal{Y}} \max\limits_{\mathbf{v} \in \mathcal{V}} \| (\mathbf{W} \odot \mathbf{Y}) \cdot \mathbf{v} \|_1$ is arbitrarily small (or zero).

**Fact 3.2.** The following two statements are equivalent:

(1) We can find a constant (independent of $n$) lower bound $\omega$ such that for any setup $(\mathbf{X}, \mathbf{W}, \mathbf{Y})$, there exists at least one vote vector and classifier $\varphi$ such that $w'_\Sigma \geq \omega$ holds.

(2) we can find a constant (independent of $n$) lower bound $\omega$ such that $\min\limits_{\mathbf{W} \in \mathcal{W}, \mathbf{Y} \in \mathcal{Y}} \max\limits_{\mathbf{v} \in \mathcal{V}} \| (\mathbf{W} \odot \mathbf{Y}) \cdot \mathbf{v} \|_1 \geq \omega$.

So, to find the lower bound $\omega$, we need to prove that $\min\limits_{\mathbf{W} \in \mathcal{W}, \mathbf{Y} \in \mathcal{Y}} \max\limits_{\mathbf{v} \in \mathcal{V}} \| (\mathbf{W} \odot \mathbf{Y}) \cdot \mathbf{v} \|_1 \geq \omega$. Let's begin with a simple $n$-dependent lower bound.

**Theorem 3.3** (An $n$-dependent lower bound). $\min\limits_{\mathbf{W} \in \mathcal{W}, \mathbf{Y} \in \mathcal{Y}} \max\limits_{\mathbf{v} \in \mathcal{V}} \| (\mathbf{W} \odot \mathbf{Y}) \cdot \mathbf{v} \|_1 \geq \frac{1}{n}$.

*Proof of Theorem 3.3.*

$$\begin{aligned}
\min_{\mathbf{W} \in \mathcal{W}, \mathbf{Y} \in \mathcal{Y}} \max_{\mathbf{v} \in \mathcal{V}} \| (\mathbf{W} \odot \mathbf{Y}) \cdot \mathbf{v} \|_1 &\overset{a}{\geq} \min_{\mathbf{W} \in \mathcal{W}, \mathbf{Y} \in \mathcal{Y}} \max_{\mathbf{v} \in \mathcal{V}} \| (\mathbf{W} \odot \mathbf{Y}) \cdot \mathbf{v} \|_\infty \\
&= \min_{\mathbf{W} \in \mathcal{W}, \mathbf{Y} \in \mathcal{Y}} \max_{\mathbf{v} \in \mathcal{V}, i \in [n]} \left| \sum_{l=1}^{K} w_{i,l} \cdot y_{i,l} \cdot v_l \right| \\
&\overset{b}{\geq} \min_{\mathbf{W} \in \mathcal{W}} \max_{i \in [n]} \sum_{l=1}^{K} w_{i,l} \overset{c}{=} \frac{1}{n},
\end{aligned}$$

where $a$ is from the fact that $\|\mathbf{x}\|_1 \geq \|\mathbf{x}\|_\infty$ where $\|\mathbf{x}\|_\infty = \max\limits_{1 \leq i \leq n} |x_i|$ is the $\ell_\infty$-norm of $\mathbf{x}$; $b$ comes from choosing $v_l = y_{i,l}$ for $l = 1, \ldots, K$ when $i$ is fixed; $c$ is from the fact that $\max\limits_{i \in [n]} \sum_{l=1}^{K} w_{i,l} \geq \frac{1}{n} \sum_{i=1}^{n} \sum_{l=1}^{K} w_{i,l} = \frac{1}{n}$ and the equation can be attained. $\qquad \square$

**Remark 1.** *The lower bound in Theorem 3.3 depends on $n$, and if we use $\frac{1}{n}$ as the lower bound of $w'_\Sigma$, then we need at most $T^* = \left\lceil \frac{2n^2 \ln(2n(K-1))}{(\delta')^2} \right\rceil + 1$ iterations to make the exponential error become zero, which quadratically increases as $n$. When the training set is large, $T^*$ becomes very large, which is one of the reasons that [14] wants to get a lower bound independent of $n$.*

Next, we introduce how we solve the open problem to get a lower bound independent of $n$.

**Theorem 3.4** (An $n$-independent lower bound). $\min\limits_{\mathbf{W}\in\mathcal{W}, \mathbf{Y}\in\mathcal{Y}} \max\limits_{\mathbf{v}\in\mathcal{V}} \|(\mathbf{W} \odot \mathbf{Y}) \cdot \mathbf{v}\|_1 \geq \frac{1}{\sqrt{2K}}$.

**Remark 2.** *Theorem 3.4 shows that there is a constant $\omega = \frac{1}{\sqrt{2K}}$ such that for any setup $\mathbf{W}, \mathbf{Y}$, there always exists a code vector $\mathbf{v}$ such that $w'_\Sigma \geq \omega$. This solves the open problem proposed by [14]. So we need at most $T^* = \left\lceil \frac{4K \ln(2n(K-1))}{(\delta')^2} \right\rceil + 1$ iterations (see Corollary 3.6) to make the exponential error become zero.*

To prove Theorem 3.4, we use the well-known Khintchine inequality [10] Lemma 3.5.

**Lemma 3.5** (10, Khintchine inequality). *Let $\{\varepsilon_n\}_{n=1}^N$ be i.i.d. random variables with $\mathbb{P}(\varepsilon_n = \pm 1) = \frac{1}{2}$ for $n = 1, \ldots, N$, i.e., a sequence with Rademacher distribution. Let $0 < p < \infty$ and let $x_1, \ldots, x_n \in \mathbb{C}$. Then*

$$A_p \left( \sum_{n=1}^N |x_n|^2 \right)^{1/2} \leq \left( \mathop{\mathbb{E}}_{\varepsilon_1, \ldots, \varepsilon_N} \left| \sum_{n=1}^N \varepsilon_n x_n \right| \right)^{1/p} \leq B_p \left( \sum_{n=1}^N |x_n|^2 \right)^{1/2}$$

*for some constants $A_p, B_p > 0$ depending only on $p$, where*

$$A_p = \begin{cases} 2^{1/2 - 1/p} & 0 < p \leq p_0 \\ 2^{1/2}(\Gamma((p+1)/2)/\sqrt{\pi})^{1/p} & p_0 < p < 2 \\ 1 & 2 \leq p < \infty \end{cases}$$

*and*

$$B_p = \begin{cases} 1 & 0 < p \leq 2 \\ 2^{1/2}(\Gamma((p+1)/2)/\sqrt{\pi})^{1/p} & 2 < p < \infty \end{cases},$$

*where $p_0 \approx 1.847$ and $\Gamma$ is the Gamma function.*

*Proof of Theorem 3.4.* The basic idea of our proof is to consider the average performance of different code vectors for fixed choices of $\mathbf{W}, \mathbf{Y}$, i.e., use the fact that the maximum is not less than the average, which gives:

$$\min_{\mathbf{W}\in\mathcal{W}, \mathbf{Y}\in\mathcal{Y}} \max_{\mathbf{v}\in\mathcal{V}} \|(\mathbf{W} \odot \mathbf{Y}) \cdot \mathbf{v}\|_1 \geq \min_{\mathbf{W}\in\mathcal{W}, \mathbf{Y}\in\mathcal{Y}} \mathop{\mathbb{E}}_{\mathbf{v}\sim D} \|(\mathbf{W} \odot \mathbf{Y}) \cdot \mathbf{v}\|_1$$

$$= \min_{\mathbf{W}\in\mathcal{W}, \mathbf{Y}\in\mathcal{Y}} \mathop{\mathbb{E}}_{\mathbf{v}\sim D} \left[ \sum_{i=1}^n \left| \sum_{l=1}^K w_{i,l} \cdot v_l \cdot y_{i,l} \right| \right]$$

for any distribution $D$ on $\mathcal{V}$.

We take $v_1, \ldots, v_K$ be independent Rademacher random variables and then get:

$$\min_{\mathbf{W}\in\mathcal{W}, \mathbf{Y}\in\mathcal{Y}} \mathop{\mathbb{E}}_{\mathbf{v}\sim D} \left[ \sum_{i=1}^n \left| \sum_{l=1}^K w_{i,l} \cdot v_l \cdot y_{i,l} \right| \right] = \min_{\mathbf{W}\in\mathcal{W}, \mathbf{Y}\in\mathcal{Y}} \mathop{\mathbb{E}}_{\varepsilon_1, \ldots, \varepsilon_K} \left[ \sum_{i=1}^n \left| \sum_{l=1}^K w_{i,l} \cdot \varepsilon_l \cdot y_{i,l} \right| \right]$$

$$\overset{a}{\geq} A_1 \min_{\mathbf{W}\in\mathcal{W}} \sum_{i=1}^n \left( \sum_{l=1}^K w_{i,l}^2 \right)^{1/2}$$

$$\overset{b}{=} \sqrt{\frac{K}{2}} \min_{\mathbf{W}\in\mathcal{W}} \sum_{i=1}^n \left( \frac{1}{K} \sum_{l=1}^K w_{i,l}^2 \right)^{1/2}$$

$$\overset{c}{\geq} \sqrt{\frac{K}{2}} \min_{\mathbf{W}\in\mathcal{W}} \frac{1}{K} \sum_{i=1}^n \sum_{l=1}^K w_{i,l}$$

$$= \frac{1}{\sqrt{2K}},$$

where $a$ applies Lemma 3.5 with $p = 1$ and the fact that $y_{i,l}^2 = 1$ for all $i, l$; $b$ puts in the value of $A_1$; $c$ uses the concavity of $\sqrt{\cdot}$ and Jensen's inequality. $\qquad\square$

With the lower bound of $w'_\Sigma$, we can now provide a lower bound of the edge $\gamma$ and convergence guarantee for the version of ADABOOST.MH proposed by [15] conditioned on the empirically $\delta'$-weak learning condition.

**Corollary 3.6** (Lower bound for $\gamma$). *If the empirically $\delta'$-weak learning condition holds, then for any* $\mathbf{X}, \mathbf{W} \in \mathcal{W}, \mathbf{Y} \in \mathcal{Y}$, *there always exists a binary classifier $\varphi^*$ and code vector $\mathbf{v}^{\max}$ such that*

$$\gamma(\mathbf{v}^{\max}, \varphi^*, \mathbf{W}) \geq \frac{\delta'}{\sqrt{2K}}.$$

*If we run ADABOOST.MH with factorized $\mathbf{h}$, then we have*

$$\widehat{R}_{EXP}(\mathbf{f}^{(T)}, \mathbf{W}) \leq \exp\left(-\frac{\delta'}{4K}T\right)$$

*and we need at most*

$$T^* = \left\lceil \frac{4K \ln(2n(K-1))}{(\delta')^2} \right\rceil + 1$$

*to make the exponential error $\widehat{R}_{EXP}(\mathbf{f}^{(T)}, \mathbf{W})$ become zero.*

*Proof of Corollary 3.6.* For any $\mathbf{W}, \mathbf{X}, \mathbf{Y}$, let $\mathbf{v}^{\max} = \arg\max_{\mathbf{v} \in \mathcal{V}} \|(\mathbf{W} \odot \mathbf{Y}) \cdot \mathbf{v}\|_1$. Let $w'_i, y'_i, w'_\Sigma$ be defined as before, where we replace $\mathbf{v}$ there by $\mathbf{v}^{\max}$. By Theorem 3.4, $w'_\Sigma \geq \frac{1}{\sqrt{2K}} > 0$.

By the empirically $\delta'$-weak learning condition, there exists a binary classifier $\varphi^*$ such that

$$\sum_{i=1}^n \frac{w'_i}{w'_\Sigma} \cdot y'_i \cdot \varphi^*(\mathbf{x}_i) \geq \delta',$$

which means that

$$\gamma(\mathbf{v}^{\max}, \varphi^*, \mathbf{W}) \geq w'_\Sigma \delta' \geq \frac{\delta'}{\sqrt{2K}}.$$

For fixed $\mathbf{W}, \mathbf{X}, \mathbf{Y}$, let $\mathbf{v}^*(\varphi)$ be the code vector depending on $\varphi$ that is defined in Equation (7). Since the choice $\mathbf{v}^*(\varphi)$ maximizes $\gamma$ when $\varphi$ is fixed, we have that:

$$\gamma(\mathbf{v}^*(\varphi^*), \varphi^*, \mathbf{W}) \geq \gamma(\mathbf{v}^{\max}, \varphi^*, \mathbf{W}) \geq \frac{\delta'}{\sqrt{2K}}.$$

Combining the arguments in Sections 2.1 and 2.2 shows $\widehat{R}_{\text{EXP}}(\mathbf{f}^{(T)}, \mathbf{W}) \leq \exp\left(-\frac{\delta'}{4K}T\right)$ and that when we run ADABOOST.MH with factorized $\mathbf{h}$, which returns $\varphi^*, \mathbf{v}^*(\varphi^*)$ at each iteration,

$$\widehat{R}_{\text{EXP}}(\mathbf{f}^{(T)}, \mathbf{W}) < \frac{1}{2n(K-1)}, \text{ i.e. } \widehat{R}_{\text{H}}(\mathbf{F}_{\mathbf{f}^{(T)}}, \mathbf{W}) = 0$$

after at most

$$T^* = \left\lceil \frac{4K \ln(2n(K-1))}{(\delta')^2} \right\rceil + 1$$

iterations. $\qquad\square$

The previous discussions are based on fixing the training set size $n$ and the number of classes $K$. Here we consider the case when they can tend to infinity. We think the reason [14] looks for a lower bound of $w'_\Sigma$ that is independent of $n$ is that the author thinks the number of examples $n$ can be arbitrarily large in some cases, which may make the lower bound of $w'_\Sigma$ arbitrarily small.

Combine our two lower bounds in Theorems 3.3 and 3.4, for any $\mathbf{X}, \mathbf{W}, \mathbf{Y}$, we can always find a $\mathbf{v} \in \{-1, +1\}^K$ such that:

$$w'_\Sigma \geq \max\left\{\frac{1}{n}, \frac{1}{\sqrt{2K}}\right\},$$

so the lower bound can become arbitrarily small only when $n$ and $K$ tend to infinity together.

# 4  Discussion

In this section, we discuss the importance of solving this problem.

In statistical learning theory, algorithms can be divided into proper and improper learning algorithms. For proper learning, the most famous algorithms are ERM [22] and its variants [26, 16, 24]. For improper learning, boosting algorithms are usually used to construct improper algorithms [2, 3, 17, 23]. Furthermore, the convergence rate of the boosting algorithm usually affects the sample complexity of the constructed algorithm, i.e., the sample complexity of the constructed algorithms usually depends on the value $T^*$ where the training error becomes zero. So boosting algorithms are basic but important tools in statistical learning theory.

In binary classification, ADABOOST [9] is one of the most famous and influential algorithms among all the binary boosting algorithms. Since the proposal of ADABOOST, many works have tried to extend the boosting framework to multi-class classification problems. Most multi-class boosting algorithms have been restricted to reducing the multi-class classification problem to multiple two-class problems, among which the most famous and influential one is ADABOOST.MH [20]. Moreover, ADABOOST.MH has inspired the proposal of many other multi-class boosting algorithms. For example, inspired by the characteristics of ADABOOST.MH that reduces the multi-class classification problem to multiple two-class problems, [13] chooses another line of thought to develop an algorithm that directly extends the ADABOOST.MH algorithm to the multi-class case without reducing it to multiple two-class problems; [1] demonstrates how to improve the efficiency and effectiveness of ADABOOST.MH and proposes the algorithm LDA-ADABOOST.MH; [18] proposes an efficient multi-class fault diagnosis approach based on the ADABOOST.MH algorithm; [7] proposes a method for ranking based on ADABOOST.MH. There are also many other works based on ADABOOST.MH [21, 12, 8]. Furthermore, many works (for example, [13, 8, 25, 27]) use ADABOOST.MH as the baseline, which further shows the importance of ADABOOST.MH. For example, the only baseline used in [13] is ADABOOST.MH. In summary, ADABOOST.MH serves as a link between binary classification boosting algorithms and multi-class classication boosting algorithms, the cornerstone of multi-class boosting, and has a big influence on the multi-class boosting field. Our work is important because it shows that Kégl's work [15], which solves the computational problem (at the level of the strong learner at least) of ADABOOST.MH, does indeed work in theory and works essentially as fast as binary ADABOOST.

# 5  Conclusion

In this paper, we resolve the open problem raised by [14] by presenting a $n$-independent lower bound for $w'_\Sigma$. In addition to that, we also provide a $n$-dependent lower bound for $w'_\Sigma$ to show that $w'_\Sigma$ may be arbitrarily small only when $n$ and $K$ tend to infinity together. Based on the lower bounds for $w'_\Sigma$ and the empirically $\delta'$-weak learning condition, we provide an upper bound for the weighted exponential error and a number $T^*$ where the weighted exponential error becomes zero after at most $T^*$ iterations.

## Acknowledgments and Disclosure of Funding

This work is supported by the National Natural Science Foundation of China under Grant 624B2106, the Key R&D Program of Hubei Province under Grant 2024BAB038, National Key R&D Program of China under Grant 2023YFC3604702, and the Fundamental Research Fund Program of LIESMARS.

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

## A  The pseudocode of the factorized ADABOOST.MH

In this section, we adapt the pseudocode of the factorized ADABOOST.MH from [14]. $\mathbf{X}$ is the $n \times d$ observation matrix, $\mathbf{Y}$ is the $n \times d$ label matrix, $\mathbf{W}$ is the user-defined weight matrix used in the definition of the weighted Hamming error (1). Let $\text{BASE}(\cdot, \cdot, \cdot)$ be the base learner algorithm, and $T$ be the number of iterations. Let $\alpha^{(t)}$ be the base coefficient $\mathbf{v}^{(t)}$ be the vote vector, $\varphi^{(t)}(\cdot)$ be the scalar base (weak) classifier, $\mathbf{h}^{(t)}(\cdot)$ be the vector-based classifier, and $\mathbf{f}^{(t)}(\cdot)$ be the final (strong) discriminant function.

---

**Algorithm 1:** The factorized ADABOOST.MH

**Input** : $\mathbf{X}, \mathbf{Y}, \mathbf{W}, \text{BASE}(\cdot, \cdot, \cdot), T$;

1   $\mathbf{W}^{(1)} = \frac{1}{n}\mathbf{W}$;

2   **for** $t \leftarrow 1$ *to* $T$ **do**

3     $\left( \alpha^{(t)}, \mathbf{v}^{(t)}, \phi^{(t)} \right) \leftarrow \text{BASE}(\mathbf{X}, \mathbf{Y}, \mathbf{W}^{(t)})$;

4     $\mathbf{h}^{(t)}(\cdot) \leftarrow \alpha^{(t)}\mathbf{v}^{(t)}\varphi^{(t)}(\cdot)$;

5     **for** $i \leftarrow 1$ *to* $n$ **do**

6       **for** $l \leftarrow 1$ *to* $K$ **do**

7         $w_{i,l}^{(t+1)} \leftarrow w_{i,l}^{(t)} \cdot \dfrac{\exp\left( -\mathbf{h}_l^{(t)}(\mathbf{x}_i)y_{i,l} \right)}{\sum_{i'=1}^n \sum_{l'=1}^K w_{i',l'}^{(t)} \exp\left( -\mathbf{h}_{l'}^{(t)}(\mathbf{x}_{i'})y_{i',l'} \right)}$;

8       **end**

9     **end**

10 **end**

**Output :** $\mathbf{f}^{(T)}(\cdot) = \sum_{t=1}^T \mathbf{h}^{(t)}(\cdot)$;

---

## B  Related Works

In addition to ADABOOST.M1, ADABOOST.M2, ADABOOST.MH, and factorized ADABOOST.MH, there are also some works on multi-class boosting.

To circumvent the hardness result for a large class of natural boosting, [5] utilizes the technique of list learning and proposes an efficient improper multi-class boosting algorithm with sample and oracle complexity bounds that are entirely independent of the number of classes.

[6] studies the resources required for boosting, especially how they depend on the number of classes $K$. [6] presents results on the sample complexity, oracle complexity, and finds a trade-off between number of oracle calls and the resources required of the weak learner.

[4] proposes an efficient multi-class boosting algorithm with the help of list learning, the success of the proposed algorithm is guaranteed by the relaxed $\gamma$-BRG condition.

In this paper, we solve the open problem proposed in [14] and provide a bound for the oracle complexity of the factorized ADABOOST.MH algorithm. The algorithm that we consider is different from those in [5, 6, 4], and the conditions are also different. We find a missing convergence result for factorized ADABOOST.MH, so we think our work is a complementary of the related works [5, 6, 4].

