# OpenReview forum: "A Boosting-Type Convergence Result for AdaBoost.MH with Factorized Multi-Class Classifiers"
_NeurIPS.cc/2024/Conference — NeurIPS 2024 poster_

### Official Review · Reviewer_HyZV · 2024-07-08

**Soundness:** 4
**Presentation:** 3
**Contribution:** 4
**Rating:** 8
**Confidence:** 4

**Summary:**

This paper studies the convergence rate of a variant of the boosting algorithm AdaBoost.MH, which is named factorized AdaBoost.MH. Factorized AdaBoost.MH replaces the one-against-all base classifier by the factorized base classifier which consists of a binary and a vote vector. This paper solves an open problem proposed in COLT 2014 and provide a convergence rate for factorized AdaBoost.MH.

**Strengths:**

- This paper solves an open problem proposed in COLT 2014, which has been open for about 10 years. The proof is clear and the basic ideas behand the proofs are given.
- This paper is well-written and easy to read.
- The background is clearly introduced, which is friendly to new readers.

**Weaknesses:**

- A minor issue: part of the presentations are similar to that in [1].


[1] Balázs Kégl. Open problem: A (missing) boosting-type convergence result for adaboost.mh with factorized multi-class classifiers. In Maria-Florina Balcan, Vitaly Feldman, and Csaba Szepesvári, editors, COLT, volume 35, pages 1268–1275, 2014.

**Questions:**

- In equation (5) of [1], the vector $\mathbf{v}$ is not assumed to belong to $\\{ -1,+1 \\}^K$, while this paper makes such an assumption. Is the setting in your paper completely the same as that in [1]?

[1] Balázs Kégl. Open problem: A (missing) boosting-type convergence result for adaboost.mh with factorized multi-class classifiers. In Maria-Florina Balcan, Vitaly Feldman, and Csaba Szepesvári, editors, COLT, volume 35, pages 1268–1275, 2014.

**Limitations:**

No.

---

> ### Author Rebuttal · Authors · 2024-07-31
>
> Thank you for your recognition. We are glad you like our paper.
>
> **For the weakness:** Thank you for your reminder. We use the presentations that are similar to that in [1] in order to make them consistent and make it convenient for the readers that have read [1] before to understand our paper. We would carefully check the presentations of our paper and change the unneccessary ones which are similar to [1].
>
> **For the question:** In fact, the vector $\mathbf{v}$ in [1] is also assumed to belong to $\\{ -1,+1 \\}^K$. The authors of [1] forget to mention it after equation (5). According to the equation (12) in [1], we know that the vector $\mathbf{v}$ considered in [1] in fact belongs to $\\{ -1,+1 \\}^K$.
>
>
> ### References
> [1] Balázs Kégl. Open problem: A (missing) boosting-type convergence result for adaboost.mh with factorized multi-class classifiers. In Maria-Florina Balcan, Vitaly Feldman, and Csaba Szepesvári, editors, COLT, volume 35, pages 1268–1275, 2014.

---

> > ### Comment · Reviewer_HyZV · 2024-08-09
> >
> > Thanks the authors for the effort in addressing my questions. I would like to keep my original recommendations.

---

> > > ### Author Response · Authors · 2024-08-10
> > >
> > > Thank you for your reply, we are glad that you are satisfied with our answers.

---

### Official Review · Reviewer_g6ZT · 2024-07-11

**Soundness:** 3
**Presentation:** 1
**Contribution:** 2
**Rating:** 4
**Confidence:** 4

**Summary:**

The paper addresses an open question posed by Kegl, 2014 regarding the convergence properties of AdaBoost.MH, a boosting algorithm designed for multi-class classification problems. Specifically, Kegl, 2014 noted that it is challenging to prove the convergence of this algorithm due to the weighted sum of binary classifications being less than one, thereby requiring a uniform lower bound on the weights. The authors of the current paper provide a solution to this open problem by establishing two convergence results: one dependent on the number of instances (n) and another independent of n, which is based on the number of classes (K).

**Strengths:**

1. The authors derive two upper bounds for the algorithm's convergence: one that depends on the number of instances (n) and another that relies solely on the number of classes (K). While the first bound may be less relevant due to its dependence on n, which can grow arbitrarily large, the second bound offers a more useful guarantee of convergence.
2. The authors further demonstrate that the lower bound on w'_{\Sigma} (the weighted sum) can become arbitrarily small only in the limiting case where both the number of instances and classes approach infinity simultaneously.

**Weaknesses:**

1.While the authors successfully address an open question in the field, I find it challenging to discern the significance of their findings beyond mere academic curiosity. As a consequence, it is unclear to me how this result contributes meaningfully to the broader machine learning community. To strengthen the case for this paper, I believe it would be beneficial for the authors to explicitly highlight the potential practical implications or applications of their convergence results, thus clarifying their relevance and value to the field.
2. Building upon my previous observation, I would like to emphasize the importance of empirical validation. While the theoretical convergence results are certainly valuable, their practical significance is amplified by experimental evidence demonstrating the correctness and effectiveness of these findings. To further strengthen the paper, I suggest that the authors include some concrete experimental results or simulations that illustrate the applicability of their proved bounds in real-world scenarios.
3. I found the writing style in the paper to be somewhat challenging to follow. The use of symbols like w'_{\Sigma} in the early sections can make it difficult for readers to quickly grasp the main ideas. Additionally, I noticed that Section 2.2 primarily recapitulates the Kegl, 2014 paper without adding significant value. Instead, I would suggest that the authors condense this information into a concise summary, allowing them to focus on presenting their own original contributions. By streamlining the writing and incorporating a clear summary of relevant background material, the authors can improve the overall readability and flow of the manuscript.

**Questions:**

1. I see that the result of Remark 2 still has ln(2n(K-1)) in the convergence result of T^{*}. It is definitely much better than the result of  Remark 1, but I am curious regarding why the authors call this particular result independent of n.

---

> ### Author Rebuttal · Authors · 2024-07-31
>
> Thank you for your suggestions. We now provide explanations about the issues in weaknesses and questions.
>
> ### **For weaknesses.**
> **For the significance of our work.** Firstly, we claim that the main contribution of our work is solving an open problem in [1] and providing a convergence rate for the factorized AdaBoost.MH algorithm, which is thought to be meaningful by the Reviewers BMi6, jA8z, and HyZV. Since this paper focuses on the theoretical perspective, we next introduce the importance of boosting algorithms from the theoretical perspective. Boosting is a very important part of statistical learning theory and AdaBoost.MH is an important algorithm in multiclass boosting. Boosting algorithms are basic but important tools in statistical learning theory, for example, in the PAC learning framework, many algorithms are constructed by boosting algorithms [2-4]. Furthermore, the convergence rate of the boosting algorithm usually affects the sample complexity of the proposed algorithm, i.e., the sample complexity of the constructed algorithms usually depends on the value $T^*$. So we believe our convergence rate can be helpful in the field of statistical learning theory.
>
> **Regarding the experimental validation.** As stated above, boosting algorithms are very useful in statistical learning theory, so our results are surely valuable, especially in applications to statistical learning theory. Since our paper is **purely theoretical**, we focus on the theoretical perspectives of the factorized AdaBoost.MH algorithm. So the experiments were beyond the scope of this work. For some results about the effectiveness of the factorized AdaBoost.MH algorithm, please refer to [5].
>
> **For the writing issues.** We are sorry for confusing you. In the early sections, we use the undefined symbol $w_ {\Sigma}^\prime$ to introduce the main concerns of [1], it is truly not suitable. In fact, the ultimate concern of [1] is a convergence rate for the factorized AdaBoost.MH algorithm, we are glad to restate the concern of [1] as an upper bound for the convergence rate rather than a lower bound for $w_ {\Sigma}^\prime$, and then the $w_ {\Sigma}^\prime$ term will not appear before its definition. Since our aim is to study the convergence rate for the factorized AdaBoost.MH algorithm, it is important to provide an overview of factorized AdaBoost.MH algorithm. We recapitulate [1] in section 2.2 to provide a complete introduction to the problem that we solve. It is a good idea to condense this information into a concise summary, we are glad to do it in our revision.
>
>
> ### **For questions.**
>
> We are sorry for confusing you. In fact, we refer to our second result as being independent of $n$ because we are talking about $w_ {\Sigma}^\prime$ (which is focused on in [1]) rather than $T^*$. In fact, the $n$ term in $T^*$ is unavoidable since we need to set the upper bound of the exponential loss to be less than $\frac{1}{2n(K-1)}$ (as done in the proof of Corollary 3.5). We will clarify it to remove ambiguity in our revision.
>
> ### **References**
>
> [1] Open problem: A (missing) boosting-type convergence result for adaboost.mh with factorized multi-class classifiers.
>
> [2] A theory of PAC learnability of partial concept classes.
>
> [3] A Characterization of Multiclass Learnability.
>
> [4] VC Classes are Adversarially Robustly Learnable, but Only Improperly.
>
> [5] The return of AdaBoost.MH: multi-class Hamming trees.

---

> > ### Comment · Reviewer_g6ZT · 2024-08-13
> > **Response to author rebuttal**
> >
> > I thank the authors for going through the review and responding to my questions. As pointed out, I am clearly the outlier in this reviewer group. I see that reviewer iRY2 also initially shared my main concern and the conversation there was very helpful. However, I feel that this significance needs to be properly mentioned in the manuscript itself which has not been done in its current format. The explanation regarding experimental validation can be defended, but some experiments should definitely elevate the quality of this paper in my opinion.
> >
> > After going through all the reviews and discussions, I have decided to improve my scores slightly.

---

> ### Author Response · Authors · 2024-08-11
> **Request for feedbacks**
>
> Dear Reviewer,
>
> Thanks for reviewing our submission.
>
> We have turned in our responses during the rebuttal period. Since it has been a few days after the beginning of the discussion period, we sincerely request for some feedback about whether we have solved your problems.

---

### Official Review · Reviewer_jA8z · 2024-07-11

**Soundness:** 3
**Presentation:** 3
**Contribution:** 4
**Rating:** 8
**Confidence:** 4

**Summary:**

The paper solves a COLT 2014 open problem. The investigated problem is about presenting a convergence rate of the factorized AdaBoost.MH algorithm, which is based on AdaBoost.MH and aims to boost weak classifiers to a strong classifier in the multiclass setting. The paper shows two lower bounds (one depends on the sample size $n$ and another depends on the class number $K$) for the important term $\omega_\Sigma$, which leads to two convergence rates for the factorized AdaBoost.MH algorithm.

**Strengths:**

**Originality.** The proofs of this paper are based on basic algebraic and probabilistic tools, so I believe the paper is original.

**Quality.** This is a high-quality paper. It is well-written and solves a COLT 2014 open problem. The proofs are sound and detailed, and the solution is elegant.

**Clarity.** The writting of this paper is good. The definitions and notations are very clear, and the conditions of the theorems and corollaries are clearly stated.

**Significance.** A long-playing open problem is beautifully solved by the paper. The paper provide a convergence rate for the effecient factorized AdaBoost.MH algorithm. So I think the paper is of great significance.

**Weaknesses:**

Firstly, the proof is a little easy compared with the solution of other open problems. However, it may not be seem as a weakness since complicated proofs are not neccessarily better.

Secondly, I think it would be better to provide some high level intuitions about idea to solve the open problem.

**Questions:**

Please refer to the weaknesses part.

**Limitations:**

No. The theoretical results are only limited to the factorized AdaBoost.MH algorithm.

---

> ### Author Rebuttal · Authors · 2024-07-31
>
> Thank you for your recognition. We are glad you like our paper.
>
> **For the first weakness:** As you say, complicated proofs are not neccessarily better. We believe that it is great to solve an open problem through a easy method.
>
> **For the second weakness:** Thank you for your reminder. The main idea of our proof is to take advantage of the properties of the terms $\mathbf{W, Y}$ and $\mathbf{v}$. The most intuitive idea is to use the property that the sum of the elements in $\mathbf{W}$ is $1$. So, in the proof, we try to get the sum of the elements in $\mathbf{W}$ or other terms that is related to the sum of the elements in $\mathbf{W}$.
> - In Theorem 3.3, we utilize the relationship between norms to get a lower bound that involves the maximum row sum of $\mathbf{W}$. Fortunately, since the sum of $\mathbf{W}$ equals $1$, we know that the maximum row sum of $\mathbf{W}$ is not less than $\frac{1}{n}$.
> - In Theorem 3.4, we consider the average performance of some different code vectors rather than the worst performance of all possible code vectors. We take $\mathbf{v}$ to be drawn from a uniform distribution on the binary cube $\\{ -1, +1 \\}^K$. And fortunately we find that the average performance can be deduced to the sum of the elements in $\mathbf{W}$ according to the Khintchine inequality and Jensen's inequality.

---

> > ### Comment · Reviewer_jA8z · 2024-08-11
> > **Keep the scores**
> >
> > I have read the response letter.

---

> > > ### Author Response · Authors · 2024-08-11
> > >
> > > Thank you for your reply. We are glad you like our paper.

---

### Official Review · Reviewer_iRY2 · 2024-07-11

**Soundness:** 3
**Presentation:** 3
**Contribution:** 3
**Rating:** 6
**Confidence:** 3

**Summary:**

This paper resolves an open problem pointed out by [7]. The open problem involves providing a lower bound, which is independent of $n$, for the coefficient $w'_{\Sigma}$ of the weighted multi-class exponential margin-based error in the factorized version of ADABOOST.MH. This result demonstrates that if the $\delta'$-weak learning condition is satisfied, the aforementioned error can be reduced to 0 by adding weak learners.
The key points of the proof are as follows:

* Reformulating the proposition to be proved as a lower bound evaluation of a minimax problem.
* Treating the code $v$ as a Rademacher random variable, bounding the maximum value from below by the expectation, and then applying Khintchine inequality.

**Strengths:**

The paper successfully resolves the open problem highlighted by [7].

**Weaknesses:**

It seems the paper does not sufficiently explain the significance of resolving the open problem pointed out by [7].

**Questions:**

* Is it possible to confirm the $\delta'$-weak learning condition is satisfied for a real-world dataset?
* If so, can we confirm how quickly the weighted multi-class exponential margin-based error decreases by numerical experiments? (Can we verify how tight $T^*$ is in practice by experiments?)

**Limitations:**

One limitation of this study appears to be the $\delta'$-weak learning condition. It would have been beneficial if the paper had provided examples or explanations to illustrate how strong this condition is for real-world datasets.

---

> ### Author Rebuttal · Authors · 2024-07-31
>
> Thank you for your suggestions. We now provide explanations about the issues in weaknesses and questions.
>
> ### **For the weakness: significance of the work.**
>
> The main contribution of our work is solving an open problem in [1] and providing a convergence rate for the factorized AdaBoost.MH algorithm, which is thought to be meaningful by the Reviewers BMi6, jA8z, and HyZV.
>
> Boosting is a very important part of statistical learning theory and AdaBoost.MH is an important algorithm in multiclass boosting. Boosting algorithms are basic but important tools in statistical learning theory, for example, in the PAC learning framework, many algorithms are constructed by boosting algorithms [2-4]. Furthermore, the **convergence rate** of the boosting algorithm usually affects the sample complexity of the proposed algorithm, i.e., the sample complexity of the constructed algorithms usually depends on the value $T^*$. So we believe our convergence rate can be helpful in the field of statistical learning theory.
>
> ### **For the questions.**
>
> Firstly, we should emphasize that the $\delta$-weak learning condition is the most basic and commonly used assumption in boosting [5]. From the above answers, we know that boosting algorithms are usually used in statistical learning theory. In such applications [2-4], they try to construct a weak learner based on some known algorithms and then apply boosting results. In such cases, the weak learners are easily to be shown to satisfy the $\delta$-weak learning condition. So, from the theoretical perspective, we can easily construct weak learners that satisfy the $\delta$-weak learning condition based on known learning algorithms, which means that **we do not need to worry about whether the $\delta$-weak learning condition holds when applying our convergence results to statistical learning theory problems**.
>
> To the best of our knowledge, there is no work focusing on validating the $\delta$-weak learning condition for real-world datasets. The validation of such a condition depends on a lot of factors, for example, it depends on the hypothesis class chosen for the weak learner, and the examples that we sample. Suppose $\gamma(\mathbf{w}, \varphi)$ is the margin of $\varphi$ under weight vector $\mathbf{w}$, and $\mathcal{H}$ be the hypothesis class used for the weak learner. By definition 2.1, to validate the empirically $\delta$-weak learning condition, we need to calculate $\underset{\mathbf{w} \in \Delta^{m-1}}{\min} \underset{\varphi \in \mathcal{H}}{\min} \gamma(\mathbf{w}, \varphi)$ and set it to be the value of $\delta$. It is obvious that calculating $\delta$ involves taking the minimum over $\mathcal{H}$ and $\Delta^{m-1}$, which is possible but generally computationally intractable.
>
> It's an interesting topic to validate the $\delta$-weak learning condition and confirm how quickly the weighted multi-class exponential margin-based error decreases by numerical experiments. However, our paper is purely theoretical, we focus on the theoretical perspectives of the factorized AdaBoost.MH algorithm. So the experiments are beyond the scope of this work.
>
> ### **References**
>
> [1] Open problem: A (missing) boosting-type convergence result for adaboost.mh with factorized multi-class classifiers.
>
> [2] A theory of PAC learnability of partial concept classes.
>
> [3] A Characterization of Multiclass Learnability.
>
> [4] VC Classes are Adversarially Robustly Learnable, but Only Improperly.
>
> [5] Boosting : Foundations and Algorithms.

---

> > ### Comment · Reviewer_iRY2 · 2024-08-10
> >
> > Thank you for your response.
> >
> > Regarding the points I raised in my questions, I was considering the possibility that adding experimental evaluations could increase the value of this paper.
> > As Reviewer g6ZT mentioned, I also believe that even for theoretical research, plotting the obtained upper and lower bounds for several examples can sometimes provide insights.
> > However, I don't intend to lower my evaluation of this paper due to the lack of experimental evaluations and your response is sufficient to me.
> >
> > Thank you also for your explanation about the importance of the research.
> > I can certainly understand that boosting is a very important technique, but could you elaborate further on your thoughts about the positioning and importance of AdaBoost.MH within the boosting methods for multi-class classification?
> >
> > For example, [1] states that AdaBoost.MH was a state-of-the-art method as of 2014, which seems to have been one of the significances of solving this problem.
> > (I'm not arguing that only state-of-the-art methods are important subjects for theoretical analysis. I'm just citing this as an example of one way to persuade readers of a problem's importance.)
> > In contrast, this paper introduces AdaBoost.M1, AdaBoost.M2, AdaBoost.MH, and AdaBoost.MH with multi-class Hamming trees.
> > However, although it mentions that AdaBoost.MH was improved by multi-class Hamming trees, there isn't much discussion about the performance of each method.
> > Also, all the research mentioned in Related Works is by the same author.
> > Are there no other studies applying boosting to multi-class classification problems?
> > For instance, the AdaBoostClassifier in scikit-learn, a widely used machine learning library, adopts the method from [2], which I believe is not cited in this paper.
> >
> > Within these various boosting methods for multi-class classification, how is AdaBoost.MH positioned practically, theoretically, or historically, and why is it important?
> > I think providing this kind of background could potentially appeal to a wider audience about the value of this important research.
> >
> > [1] B. Kégl, Open Problem: A (Missing) Boosting-type Convergence Result for AdaBoost.MH with Factorized Multi-class Classifiers, Proceedings of The 27th Conference on Learning Theory, PMLR 35:1268-1275, 2014.
> > [2] J. Zhu, H. Zou, S. Rosset, T. Hastie, “Multi-class adaboost.” Statistics and its Interface 2.3 (2009): 349-360.

---

> > > ### Author Response · Authors · 2024-08-11
> > >
> > > Thank you for your reminder, we are glad to add more discussions on the importance of AdaBoost.MH in our revision.
> > >
> > > In binary classification, AdaBoost [1] is one of the most famous and influential algorithms among all the binary boosting algorithms. The paper [1], which proposes AdaBoost, has more than 22000 citations. Since the proposal of AdaBoost, there are many works trying to extend the boosting framework to multi-class classification problems. Most multi-class boosting algorithms have been restricted to reducing the multi-class classification problem to multiple two-class problems, among which the most famous and influential one is AdaBoost.MH [2]. The paper [2], which proposes AdaBoost.MH, has more than 4900 citations. Moreover, AdaBoost.MH has inspired the proposal of many other multi-class boosting algorithms. For example, inspired by the characteristics of AdaBoost.MH that reduces the multi-class classification problem to multiple two-class problems, [3] chooses another line of thought to develop an algorithm that directly extends the AdaBoost algorithm to the multi-class case without reducing it to multiple two-class problems; [4] demonstrates how to improve the efficiency and effectiveness of AdaBoost.MH and proposes the algorithm LDA-AdaBoost.MH; [5] proposes an efficient multi-calss fault diagnosis approach based on the AdaBoost.MH algorithm; [6] proposes a method for ranking based on AdaBoost.MH. There are also many other works based on AdaBoost.MH [7-9]. Furthermore, to the best of our knowledge, many works (for example, [3, 9-11]) use AdaBoost.MH as the baseline, which further shows the importance of AdaBoost.MH. For example, the only baseline used in [3] is AdaBoost.MH. In summary, AdaBoost.MH serves as a link between binary classification boosting algorithms and multi-class classification boosting algorithms, the cornerstone of multi-class boosting, and has a big influence on the multi-class boosting field.
> > >
> > > From the application perspective, AdaBoost.MH shows influence on ranking [6],  computer vision [10], natural language processing [4, 9]. multimedia [8]. Moreover, AdaBoost.MH is also realized by a boosting package MultiBoost [12].
> > >
> > > We are glad to add the above discussions in our revision. If you have further questions, please fell free to raise them and we are glad to discuss with you.
> > >
> > >
> > > ### **References**
> > >
> > > [1] A decision theoretic generalization of on-line learning and an application to boosting.
> > >
> > > [2] Improved Boosting Algorithms Using Confidence-rated Predictions.
> > >
> > > [3] Multi-class AdaBoost.
> > >
> > > [4] LDA-AdaBoost.MH: Accelerated AdaBoost.MH based on latent Dirichlet allocation for text categorization.
> > >
> > > [5] An Effective Fault Diagnosis Approach Based On Gentle AdaBoost and AdaBoost.MH.
> > >
> > > [6] Ranking by calibrated AdaBoost.
> > >
> > > [7] An improved boosting algorithm and its application to text categorization.
> > >
> > > [8] Generalized multiclass adaboost and its applications to multimedia classification.
> > >
> > > [9] MP-Boost: A Multiple-Pivot Boosting Algorithm and Its Application to Text Categorization.
> > >
> > > [10] Improved multiclass AdaBoost for image classification: The role of tree optimization.
> > >
> > > [11] A New Multiclass Generalization of AdaBoost.
> > >
> > > [12] MultiBoost: a multi-purpose boosting package.

---

> > > > ### Comment · Reviewer_iRY2 · 2024-08-12
> > > >
> > > > Thank you very much for your detailed explanation.
> > > > My concerns have been addressed.
> > > > I would like to improve my score.

---

> > > > > ### Author Response · Authors · 2024-08-12
> > > > >
> > > > > Thank you for improving the score, we are glad that we have addressed your concerns.

---

### Official Review · Reviewer_BMi6 · 2024-07-12

**Soundness:** 4
**Presentation:** 4
**Contribution:** 4
**Rating:** 9
**Confidence:** 4

**Summary:**

Balázs presented a factorized version of ADABOOST.MH and empirically show that this extension achieves promising results. Balázs further raises the convergence property of factorized ADABOOST.MH as an open problem. This submission addresses this open problem and presents an elegant proof.

**Strengths:**

This submission did a really good job in presenting the stories of ADABOOST, ADABOOST.MH and the factorized version of ADABOOST.MH. The backgrounds are also clearly shown in the paper, which makes the researchers who are not familiar with this kind of topic can fast gain the main ideas of this paper. Actually, I enjoy the reading of this paper. Technically, it is interesting to transform the open problem to a minmax problem, and both n-independent lower bound and n-dependent lower bound are provided in this paper. The Khintchine inequality plays the key role in the proof. The proof is simple and easy to follow. The significance of this paper is very important, because it solves a long lasting open problem.

**Weaknesses:**

I did not find the significant weakness about this paper. But I still have some questions. I hope the author can address them.
1, Why there is the absolute value in the equations between lines 217 -218?
2, In Corollary 3.5, why the exponential error can be zero? Is this an asymptotic result or non- asymptotic result?
3, What are the derivations between lines 232 and 233? Can you provide more details?
4, Why the first equation below line 219 hold? Can you explain this?

**Questions:**

See above.

**Limitations:**

The authors answers NA in the Checklist.

---

> ### Author Rebuttal · Authors · 2024-07-31
>
> Thank you for your recognition. We are glad you like our paper. Next, we answer your questions.
>
> 1. **Why there is the absolute value in the equations between lines 217 -218?** Answer: Because we are handling the $\ell_ 1$-norm of a vector, and the $\ell_ 1$-norm of a vector is the sum of the absolute value of its elements.
> 2. **In Corollary 3.5, why the exponential error can be zero? Is this an asymptotic result or non-asymptotic result?** Answer: We are sorry for confusing you, in fact it is a typo, we want to say that the **Hamming loss** can be zero. The reason why the Hamming loss becomes zero after $T^*$ steps is shown in the lines 237-238. It is a non-asymptotic result since we explicitly provide the convergence rate that is related to $T, K, \delta^\prime$.
> 3. **What are the derivations between lines 232 and 233? Can you provide more details?** Answer: For the given $\mathbf{W,X,Y}$, by Theorem 3.4, there exists $\mathbf{v}^{\max}$ such that $w_ {\Sigma}^\prime \ge \frac{1}{\sqrt{2K}} > 0$. Since $w_ {\Sigma}^\prime > 0$, we know that $\gamma\left( \mathbf{v}^{\max} , \varphi^*, \mathbf{W} \right) = \sum_ {i=1}^n w_ i^\prime \cdot y_ i^\prime \cdot \varphi^*(\mathbf{x}_ i) = w_ \Sigma^\prime \cdot \sum_ {i=1}^n \frac{w_ i^\prime}{w_ \Sigma^\prime} \cdot y_ i^\prime \cdot \varphi^*(\mathbf{x}_ i) \ge w_ \Sigma^\prime \delta^\prime \ge \frac{\delta^\prime}{\sqrt{2K}}$. The main skill of this step is to construct weights $\frac{w_ i^\prime}{w_ \Sigma^\prime}$ that form a distribution, the construction is dependent on the fact that $w_ \Sigma^\prime > 0$.
> 4. **Why the first equation below line 219 hold? Can you explain this?** Answer: In this line, we consider choosing $\mathbf{v}$ to be the Rademacher random vector with **independent** elements. By the independence, we can just replace the random variables $v_ 1, \dots, v_ K$ with $K$ independent Rademacher random variables $\varepsilon_ 1, \dots, \varepsilon_ K$.

---

> > ### Comment · Reviewer_BMi6 · 2024-08-10
> > **response**
> >
> > The author did a good job in answering my questions. I am happy to recommend the acceptance.

---

> > > ### Author Response · Authors · 2024-08-10
> > >
> > > Thank you for your reply, we are glad that you are satisfied with our answers.

---

### Decision · Program_Chairs · 2024-09-25

**Decision:**

Accept (poster)

**Comment:**

When it comes to bringing boosting to multiclass problems, the easiest-to-go path is usually indeed by a reduction to as many binary classification problems as required. Among others, one downside of this approach is that it is computationally demanding: instead of $T$ weak classifiers, one has to train $K T$ of them.

All but one reviewers concur with the assessment that what the paper proposes is an important result. I have read the paper and agree to an extent. It is important because it shows that Kegl's approach, which solves the computational problem (at the level of the strong learner at least), does indeed work in theory, and works essentially as fast as binary AdaBoost. The proof is simple and elegant, the sort of characteristics that have been observed elsewhere in famous boosting papers. Simplicity works in favor of the paper. I also like the dependency in L245: it makes sense that in *some* cases, the lowerbound does have to depend on $n$ (if the number of classes is very large).

The only reason why I do not rejoin my fellow reviewers on their commendations is because the proof of Theorem 3.4 is non-constructive. This could partially explain why the authors have been reluctant to provide experiments. A constructive proof with a simple, low complexity algorithm would have sealed the problem -- not just the open problem, but "the" multiclass boosting problem itself. I can see at least two potential algorithms following either from the presentation of the problem ("LP-style") or the proof technique (randomized) but each has downside. If there is something that boosting has taught, it is the value of simplicity and originality so there perhaps exists a simple original trick that would do the job. It would be good for the authors to elaborate on this (no additional result, just a discussion), as well as some of of the fixes suggested by reviewers.

I also do not consider that experiments would have been necessary for such a paper to be accepted. In conclusion, it is a fine paper which deserves to be published.

In the camera-ready version, I *strongly* encourage the authors to do some good polish of their paper, in particular number some important (in)equalities, polish the text to mix the story-telling style adopted with a more conventional approach. In particular, with the +1 camera ready page, add a section on why this problem is important (state of the art after introduction on both the technical and practical side of the problem, discussion at the end with references). Consider it as a necessary step to get a polished camera-ready that would also include readers with the same expectation as g6ZT. This will only boost further the paper's reading basis.